# At Mean 30-Year Follow-Up, Cervical Spine Disease Is Common and Associated with Thoracic Hypokyphosis after Pediatric Treatment of Adolescent Idiopathic Scoliosis

**DOI:** 10.3390/jcm11206064

**Published:** 2022-10-14

**Authors:** Ernest Young, Christina Regan, Bradford L. Currier, Michael J. Yaszemski, A. Noelle Larson

**Affiliations:** Department of Orthopedic Surgery, Mayo Clinic, Rochester, MN 55905, USA

**Keywords:** adolescent idiopathic scoliosis, hypokyphosis, cervical spine degeneration

## Abstract

Patients with adolescent idiopathic scoliosis (AIS) often have reduced sagittal thoracic kyphosis (hypokyphosis) and cervical lordosis causing an uneven distribution of physiologic load. However, the long-term consequences of hypokyphosis in AIS patients have not been previously documented. To evaluate whether uneven load distribution leads to future complications in patients with AIS, we conducted a retrospective chart review and subsequently surveyed 180 patients treated for idiopathic scoliosis between 1975 and 1992. These patients all had a minimum follow-up time of 20 years since their treatment. We observed a ten-fold increase in the incidence of anterior cervical discectomy and fusion (ACDF) compared to reported rates in the non-pathologic population. Out of the 180 patients, 33 patients met the criteria and returned for follow-up radiographs. This population demonstrated a statistically significant increased rate of cervical osteoarthritis and disc degeneration. Overall, our study suggests that hypokyphosis in patients with AIS presents with increased rates of cervical spine degeneration and dysfunction, suggesting that these patients may require additional follow-up and treatment.

## 1. Introduction

The spinal architecture with its reciprocating sagittal curves allows for physiological load absorption and maintenance of an upright posture. Sagittal plane alignment of the thoracolumbar spine has been shown to affect clinical outcomes [1,2,3,4], and appropriate cervical spine sagittal alignment has increasingly been reported as an important factor affecting postsurgical outcomes, quality of life and rates of cervical spondylosis [5,6,7,8].

Patients with adolescent idiopathic scoliosis (AIS) typically have reduced sagittal thoracic kyphosis (hypokyphosis) and decreased cervical lordosis [9,10,11,12]. The loss of kyphosis in the thoracic spine may translate into compensatory kyphosis in the cervical and lumbar spine [9,13]. If the sagittal plane is not corrected, scoliosis surgery may perpetuate thoracic hypokyphosis, driving the cervical spine into further compensatory kyphosis [10,11,14,15,16]. Loss of cervical lordosis in the adult population has been associated with increased cervical osteoarthritis, increasing deformity, pain and debility [17,18,19]. Rates of cervical spine degenerative changes and surgery for patients with adolescent idiopathic scoliosis have not been thoroughly explored in the literature.

Thus, we sought to evaluate cervical spine alignment, degenerative osteoarthritis, and the need for subsequent cervical spine surgery in a population of AIS patients with long-term follow-up. We hypothesized that thoracic hypokyphosis would be associated with degenerative changes in the cervical spine.

## 2. Materials and Methods

Between 1975 and 1992, a total of 2661 patients were treated for idiopathic scoliosis at a tertiary referral center. Of these 2661 patients, 733 were aged 0–21 years at the time of their initial scoliosis treatment. Upon medical record review, a total of 344 patients were confirmed with the diagnosis of AIS with childhood curves greater than 35 degrees treated with observation, bracing, or surgery. Patients with associated syndromes, neurologic abnormalities, infantile- or juvenile-onset, or congenital scoliosis were excluded from the study. Only idiopathic scoliosis patients seen before adulthood with appropriate radiographs and medical documentation were included. Medical records and radiographs were reviewed to identify all qualifying patients, demographics, and treatment protocols.

From the cohort of 344 eligible patients, 2 had died–one of unexplained pulmonary failure. A standardized survey was mailed to patients, including requests to describe additional spine or chest wall surgery and a battery of patient-reported outcome measures including Scoliosis Research Society scores, Spinal Appearance Questionnaires, Oswestry Disability Index, and EQ5D. Responses were received from 180 patients (52%). Mean follow-up was 30 years (range, 20–37).

In order to accurately assess sagittal alignment of the spine over time, the study only included patients that had high quality anteroposterior and lateral radiographs following scoliosis treatment at skeletal maturity (Risser 4 or 5) and at final follow-up. Radiographs taken at skeletal maturity for patients that had surgery were postoperative. A total of 33 patients met these criteria and returned for evaluation and follow-up radiographs either at the study center or locally. Childhood treatment included bracing (9), observation (3), and spinal fusion (21). Nonoperative patients (12) were compared to those undergoing childhood operative fusion (21).

Braced patients were treated in a Milwaukee brace. Surgery was typically performed for curves above 45–50 degrees and involved posterior fusion and instrumentation (*n* = 19) or anterior fusion and instrumentation (*n* = 2). Posterior instrumentation included 1st and 2nd generation Harrington rods (12) or hooks (7). Patients treated with 1st generation Harrington rods typically had pre- and postoperative casting and/or bracing.

Radiographs at skeletal maturity and most recent follow-up were compared to evaluate both coronal and sagittal curve progression as well as degenerative changes over the follow up period. The childhood radiographs of the patients had their respective curves graded per the Lenke classification with appropriate lumbar modifier [20]. Coronal thoracic and lumbar Cobb angles at skeletal maturity and at most recent follow up were evaluated for changes. The following sagittal spinal parameters were assessed: (1) C2–C7 lordosis angle (CL), (2) C2–C7 sagittal vertical axis (SVA) (distance between the C2 plumb line and C7), (3) C7 SVA (distance between C7 plumb line and the posterior superior corner of S1), (4) thoracic kyphosis (TK) (angle between T1 and T12), and (5) thoracic slope (angle made between the superior endplate of T1 and the horizontal).

The degree of cervical spine spondylosis was measured on lateral radiographs at final follow up using the commonly used grading system created by Kellgren and Lawrence [21,22,23,24]. Both disc space narrowing and osteophytes were graded on a score of 0 to 3 as follows: 0—no disease, 1—mild disease, 2—moderate disease and 3—severe disease with disc narrowing with anterior and posterior osteophytes. Cervical disc spaces 2–3 through 6–7 were utilized for this grading. For the purpose of statistics the gross sum of all grades across the 5 cervical disc spaces were used. For categorization analysis, the degree of cervical osteoarthritis per patient was defined as the maximum score across the disc segments.

For all sagittal alignments, a positive reading denotes kyphosis and negative lordosis. Cervical sagittal alignment was split into three groups based on Cobb Angle for comparison: Kyphosis (>10°), Straight or normal (10° to −10°) and lordotic (<−10°). Thoracic alignment was also split into three groups for comparison: hypokyphotic (<20°), normokyphotic (20°–40°), hyperkyphotic (>40°).

Continuous variables were assessed using a student’s two-tailed t-test, and dichotomous variables were analyzed using a Chi-square analysis. Due to the small sample size, multivariate analysis was not undertaken. Statistical significance was set at *p* < 0.05.

## 3. Results

Of the 180 patients who responded to the survey, four had undergone a cervical spine procedure during the follow-up period. These procedures included single-level anterior cervical discectomy (ACDF) (2), multilevel ACDF (1), and discectomy (1) (2.2%). One additional patient had cervical radiculopathy and severe degenerative changes. Reported rates for ACDF in the general adult population are 0.03–0.16%. Thus, our population had a ten-fold higher rate of ACDF than expected.

The cohort with adequate radiographs both at skeletal maturity and latest follow-up consisted of 30 women and 3 men with an average age at follow-up of 42.6 ± 6.5 years. The majority of patients had inadequate childhood radiographs that either did not include the cervical spine or had poor image quality for the cervical spine. The average time from radiographs at skeletal maturity to final follow-up was 28.3 ± 5.4 years. There was no difference overall in the sagittal cervical and thoracic alignment between the nonoperative and operative cohorts (Table 1).

Of these 33 patients in our cohort, 6 patients (18.8%) had additional thoracolumbar spinal surgery and 2 (6.2%) others had cervical spine surgery (one patient underwent C4–C6 anterior fusion and the other underwent C4–C5 anterior fusion). A total of 12 of the 33 (36%) had recently seen a chiropractor or pain specialist for back pain, and 4 of the 33 (9%) had consulted a specialist for neck pain.

Using Kellgren–Lawrence scoring, we found that 58% of the cohort (19/33) had moderate or severe cervical osteoarthritis (Figure 1). Those with minimal cervical osteoarthritis had a mean age of 40 ± 4.2 years, moderate had a mean age of 39 ± 5.8, and severe cervical osteo arthritis were 50 ± 3.2 years (*p* < 0.01). There was no difference in the degree of cervical osteoarthritis between the surgical and nonoperative cohorts (Figure 1).

Hypokyphosis of the thoracic spine at skeletal maturity and at latest follow-up compared to normokyphosis or hyperkyphosis correlated with increased adult cervical disc degeneration (*p* = 0.028 and 0.0002, respectively) (Table 2). Cervical osteoarthritis was increased in patients with hypokyphosis at latest follow-up compared to patients with normokyphosis or hyperkyphosis (*p* = 0.0085). There was no correlation between the degree of osteoarthritis or disc degeneration versus cervical sagittal alignment at skeletal maturity or at final follow-up.

Sagittal alignment measurements were compared for radiographs taken at skeletal maturity and at final follow-up. There were no significant differences in C7 SVA, C2C7SVA, T1 Slope and cervical or thoracic sagittal alignment between the patient’s radiographs as adolescents and those at follow up (*p* > 0.5, Table 3). Additionally, there was no significant difference between these measurements taken in adolescence and at adulthood when looking at the operative and non-operative patients separately. However, lumbar and pelvic parameters were found to change significantly, including decreased lumbar lordosis, increased pelvic tilt and resultant decreased sacral slope as these two parameters vary inversely (Table 3). Pelvic incidence did not change significantly as expected.

Analysis of cervical sagittal alignment at follow up found that the amount of the cervical kyphosis did not correlate with thoracic kyphosis or any other spine alignment variable including pelvic parameters (*p* > 0.5). This held true when looking at the non-operative vs. operative patients.

The data on the patient reported outcome measures (SRS, ODI and EQ5D) did not show any difference based on cervical sagittal alignment. However, the thoracic alignment at follow up was associated with both SRS Function and ODI. For patients with thoracic hypokyphosis, the ODI was 28.7 ± 18.6 in comparison to the patients with normal kyphosis with an ODI of 8.6 ± 8.6 (*p* = 0.008).

## 4. Discussion

In the adult spinal deformity literature, cervical kyphosis has been associated with increased degenerative disease and pain [7,8,18]. Cervical sagittal plane alignment after surgery for adolescent idiopathic scoliosis has become an increasingly important topic [9,10,11,14,15,16].

However, the long-term effects of cervical kyphosis in patients with adolescent idiopathic scoliosis are not well defined. This study reviewed a unique cohort of patients with a minimum of 20 years follow up who had undergone childhood treatment for AIS. In particular, we sought to assess the association between cervical sagittal alignment, cervical osteoarthritis, cervical spine surgery, and patient related outcomes.

The prevalence of cervical arthritic changes has been detailed in the literature [23,24,25,26,27] and has been reported from 25–46% in people aged 40–49 and 65–73% in people aged 50–59 [24,25]. The actual rate of moderate to severe cervical osteoarthritis and progression of disease in the 40–50 age group has been reported from 16–20% [23,26]. In comparison, our study cohort had a prevalence of 30% moderate degeneration and 28% severe degeneration with a combined prevalence of 58%. Additionally, the majority of the patients were in the 40–50 age range (81%) with only 5 patients (15%) being above the age of 50. Thus, our cohort may have more advanced cervical osteoarthritis than the average population, although a matched cohort of patients without scoliosis would provide more reliable comparison data.

Two (6.2%) of the patients had undergone cervical spine surgery, which is much higher than the reported incidence of anterior spinal fusion in the general population (0.05–0.2%) [28,29]. From our survey cohort, four out of 180 patients (2.2%) reported undergoing cervical spine surgery in adulthood. Interestingly, only 9% of our cohort complained of cervical pain. Edgar et al. in 1988 followed up patients with AIS who had been observed or had posterior spinal fusion at minimum of 10 years from skeletal maturity and found that neck pain was more prevalent (17.6%) in the patients who had fusion versus those who did not (7.8) [30]. Our cohort’s prevalence of 9% falls within this range but did not show a difference between those patients who were treated nonoperatively (1/12 patients, 8.3%) compared to those who underwent fusion surgery (2/21 patients, 9.5%).

The cervical alignment in patients with AIS has been discussed broadly in the literature as it differs from that of the general population. The natural tendency of the cervical spine in the general population is toward lordosis ranging from 15 to 40 degrees of lordosis with only 9% prevalence of kyphosis [7,26,27]. In AIS, there is a high prevalence of cervical kyphosis which ranges from 34 to 54% [10,11,13,14,15,16]. This number increases to as high as 89% of patients if the “straight” cervical profiles are included [9]. Our cohort showed an overall prevalence of 48% kyphosis (>10 degrees) and 21% “straight” cervical profile. This phenomenon is postulated to be caused by the three-dimensional deformity of AIS, and a compensatory cervical kyphosis develops secondary to the loss of thoracic kyphosis seen AIS. This has been shown in recent studies evaluating the effect of surgery on the cervical alignment in patients with AIS. Persistent hypokyphosis of the thoracic spine after spinal fusion surgery is thought to result in persistent postoperative cervical spine kyphosis [10,11,14,15]. In our study, however, there was no correlation between cervical and thoracic sagittal plane alignment. This may be secondary to the use of only postoperative films in comparison or small sample size.

Cervical kyphosis is often maligned in the literature as a cause of pain and degeneration in the adult cervical spine postoperative patient [5,18,31]. These patients however already have pathology or deformity, and studies addressing only healthy populations have not shown that cervical alignment is associated with increased symptoms or degeneration [7]. Interestingly, we did show that cervical degeneration at follow-up was correlated with the thoracic alignment at skeletal maturity. Thus, increased thoracic spine hypokyphosis was associated with increased cervical degeneration, perhaps due to compensatory changes in the cervical spine.

The limitations of this study include the low number of patients in the cohort. Most of the identified patients had to be excluded due to inadequate imaging of the cervical spine. This may under power the study, especially considering that it includes both operative and nonoperative patients. However, patient numbers in this study are similar to other short-term follow-up studies on cervical alignment in AIS. Similarly, most long-term follow-up studies are for a minimum of 10 years or have numbers similar to our study [32,33,34,35,36]. Further, lateral childhood radiographs were taken of the entire spine and thus are subject to parallax. The majority of current lateral radiographs of the entire spine were taken using the EOS, which eliminates parallax and has shown to provide good inter- and intra-observer reliability of cervical spine measurements in scoliosis patients. Further, flexion-extension radiographs were not available. Quality of the childhood radiographs compared to adult radiographs may be variable and may alter radiographic measurements. Patients undergoing nonoperative management less frequently had lateral scoliosis radiographs available for analysis; thus, there were fewer nonoperative patients who could be included in our series. Patients who returned for radiographs may not be representative of the entire cohort. Lastly, this study distributed both thoracic and cervical alignment into separate groups by defining both lordosis and kyphosis using set groupings. This was mitigated using the quantitative variables in these settings in order to confirm any findings in which these groupings were used. This is also reflective of the literature which does not have a set definition of both cervical kyphosis and thoracic hypokyphosis.

In summary, this study provides long-term evaluation of the cervical spine after previous treatment of AIS. Cervical spine degeneration is common in these patients and is likely more severe than patients of similar age without scoliosis. Furthermore, our AIS population had a higher rate of cervical spine surgery in adulthood than expected from other population reports. Both cervical and sagittal alignment did not change with age or treatment, although there was a higher rate of degenerative changes in patients with decreased thoracic kyphosis.

## Figures and Tables

**Figure 1 jcm-11-06064-f001:**
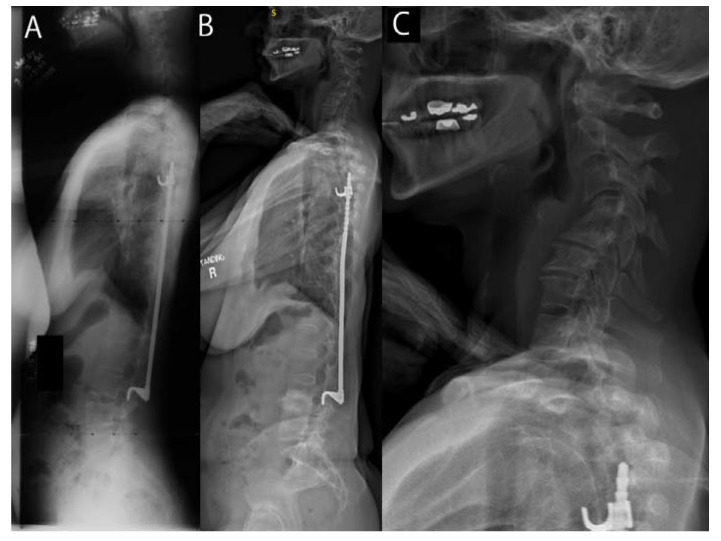
(**A**) Child treated with Harrington instrumentation for AIS at age 16. (**B**,**C**) Now at 51, she has progressive cervical kyphosis, with severe radicular pain and degenerative cervical arthritis.

**Table 1 jcm-11-06064-t001:** Comparison of nonoperative and operative cohort at follow up.

	Nonoperative (*n* = 12)	Operative (*n* = 21)	*p*-Value
Gender (Female/Male)	10/2	21/0	0.05
Age at Last Childhood Radiograph (Years)	17.0 ± 17	16.4 ± 2.6	0.50
Risser at Last Childhood Radiograph	4.5 ± 0.5	4.5 ± 0.6	0.81
Age at Follow-up (Years)	42.1 ± 1.8	43.5 ± 1.3	0.53
Time to Follow-up (Years)	28.7 ± 5.5	28.0 ± 5.5	0.75
Lenke Curve Type	1–4 2–1 3–3 5–4	1–9 2–2 3–7 5–3	
Lumbar Modifier	A—5, B—0, C—7	A—8, B—2, C—11	
Cervical Alignment Group at Follow up	5—Kyphotic 2—Straight 4—Lordotic	10—Kyphotic 6—Straight 5—Lordotic	0.44
Cervical Kyphosis at Follow up (degrees)	1.96 ± 10.2	3.8 ± 14.8	0.71
Thoracic Kyphosis Grouping at Follow up	1 Hypokyphosis 8 Normokyphosis 3 Hyperkyphosis	4 Hypokyphosis 11 Normokyphosis 6 Hyperkyphosis	0.6424
Thoracic Kyphosis at Follow up (degrees)	32.9 ± 9.3	31.3 ± 16.7	0.77

**Table 2 jcm-11-06064-t002:** Comparison of cervical and thoracic sagittal alignment and cervical degeneration at skeletal maturity and at final follow-up. comparison of cervical and thoracic sagittal alignment and cervical degeneration at final follow-up.

Skeletal Maturity				
	Thoracic Hypokyphosis (*n* = 4)	Thoracic Normokyphosis (*n* = 20)	Thoracic Hyperkyphosis (*n* = 9)	*p* Value
Cervical Osteoarthritis	2.5 ± 0.40	1.65 ± 0.18	2.0 ± 0.27	0.14
Cervical Disc Degeneration	1.25 ± 0.21	0.59 ± 0.09	0.73 ± 0.15	0.028 *
Skeletal Maturity				
	Cervical Hypokyphosis (*n* = 14)	Straight Cervical Alignment (*n* = 6)	Cervical Lordosis (*n* = 13)	*p* value
Cervical Osteoarthritis	1.79 ± 0.22	2.33 ± 0.34	1.56 ± 0.28	0.22
Cervical Disc Degeneration	0.69 ± 0.13	0.93 ± 0.20	0.51 ± 0.16	0.26
Final Follow-Up				
	Thoracic Hypokyphosis (*n* = 5)	Thoracic Normokyphosis (*n* = 19)	Thoracic Hyperkyphosis (*n* = 9)	*p* value
Cervical Osteoarthritis	2.8 ± 0.33	1.79 ± 0.17	1.44 ± 0.24	0.009 *
Cervical Disc Degeneration	1.28 ± 0.16	0.73 ± 0.08	0.30 ± 0.13	0.0002 *
Final Follow-Up				
	Cervical Hypokyphosis (*n* = 14)	Straight Cervical Alignment (*n* = 11)	Cervical Lordosis (*n* = 8)	*p* value
Cervical Osteoarthritis	2.0 ± 0.23	1.73 ± 0.26	1.75 ± 0.30	0.68
Cervical Disc Degeneration	0.85 ± 0.13	0.64 ± 0.14	0.58 ± 0.16	0.36

* denotes statistical significance.

**Table 3 jcm-11-06064-t003:** Comparison of sagittal measurements between skeletal maturity and at final follow up.

	Skeletal Maturity	Adulthood Follow Up	*p* Value
Cervical Kyphosis	3.1 ± 13.2	0.8 ± 13.5	0.5
Thoracic Kyphosis	29.2 ± 13.0	31.9 ± 14.3	0.41
T1 Slope	17.8 ± 7.9	18.6 ± 10.4	0.78
C2C7SVA	13.3 ± 12.7	9.6 ± 19.3	0.38
C7 SVA	2.6 ± 26.9	−6.6 ± 38.3	0.28
Lumbar Lordosis	43.1 ± 14.2	37.1 ± 17.5	0.02 *
Pelvic Incidence	53.8 ± 12.5	57.4 ± 11.9	0.12
Pelvic Tilt	12.9 ± 8.7	20.6 ± 9.2	0.001 *
Sacral Slope	41.1 ± 8.5	36.9 ± 14.1	0.01 *

* denotes statistical significance.

## Data Availability

Not applicable.

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
