# Peer review of "At Mean 30-Year Follow-Up, Cervical Spine Disease Is Common and Associated with Thoracic Hypokyphosis after Pediatric Treatment of Adolescent Idiopathic Scoliosis"

_jcm, 2022, doi:10.3390/jcm11206064_

Round 1

Reviewer 1 Report

This topic is very interesting, look at these points to improve the manuscript:

- Lines 21-23: "our study found that hypokyphosis in patients with AIS presents with increased rates of cervical spine degeneration... treatment." This is the conclusion, but the title does not seem to reflect this conclusion. Please revise.

- Lines 43-44: "The overarching goal of this research effort is to determine whether sagittal plane alignment affects long-term cervical spine outcomes in AIS patients" Please improve. What is the aim of this paper?

- Lines 109-110: "The cohort with adequate radiographs both at skeletal maturity and latest follow-up consisted of 30 women and 3 men". So only 33 patients, why were other patients excluded? This is not clear and must be revised.

- Lines 212-214: "Cervical kyphosis is often maligned in literature as a cause of pain and degeneration in the adult cervical spine postoperative patient" discuss more. As also in trauma. Look at these recent papers: -- doi: 10.1097/BRS.0000000000004429 -- doi: 10.3171/2022.6.SPINE22454

- Lines 152-154: "However, lumbar and pelvic parameters were found to change significantly, including lumbar lordosis, pelvic tilt and sacral slope (Table 3). Pelvic incidence did not change significantly"  - As PI did not change significantly, PT and SS change in an inverse proportional way. Discuss more these results. What do author think about these results?

- Figure 1 did not make much sense. Why did authors report this single case? It can also be removed.

Author Response

  • Lines 21-23: "our study found that hypokyphosis in patients with AIS presents with increased rates of cervical spine degeneration... treatment." This is the conclusion, but the title does not seem to reflect this conclusion. Please revise.

Agreed.  We have changed title to:  

At Mean 30-Year Follow-Up, Cervical Spine Disease Is Common and Associated with Thoracic Hypokyphosis after Pediatric Treatment of Adolescent Idiopathic Scoliosis

  • Lines 43-44: "The overarching goal of this research effort is to determine whether sagittal plane alignment affects long-term cervical spine outcomes in AIS patients" Please improve. What is the aim of this paper?

Agreed.  We have added this hypothesis:

We hypothesized that thoracic hypokyphosis would be associated with degenerative changes in the cervical spine. 

  • Lines 109-110: "The cohort with adequate radiographs both at skeletal maturity and latest follow-up consisted of 30 women and 3 men". So only 33 patients, why were other patients excluded? This is not clear and must be revised.

Unfortunately, the childhood x-rays frequently did not include the c-spine, and these patients were excluded.  We have added this to the manuscript "The majority of patients had inadequate childhood radiographs that either did not include the cervical spine or had poor image quality for the cervical spine."

Also in the limitations section, "Most of the identified patients had to be excluded due to inadequate imaging of the cervical spine."   

- Lines 212-214: "Cervical kyphosis is often maligned in literature as a cause of pain and degeneration in the adult cervical spine postoperative patient" discuss more. As also in trauma. Look at these recent papers: -- doi: 10.1097/BRS.0000000000004429 -- doi: 10.3171/2022.6.SPINE22454

Thank you for these important references.  We have reviewed them.

  • Lines 152-154: "However, lumbar and pelvic parameters were found to change significantly, including lumbar lordosis, pelvic tilt and sacral slope (Table 3). Pelvic incidence did not change significantly"  - As PI did not change significantly, PT and SS change in an inverse proportional way. Discuss more these results. What do author think about these results?
  • We have added additional discussion to this topic.  "However, lumbar and pelvic parameters were found to change significantly, including decreased lumbar lordosis, increased pelvic tilt and resultant decreased sacral slope as these two parameters vary inversely (Table 3). Pelvic incidence did not change significantly as expected."
  • Figure 1 did not make much sense. Why did authors report this single case? It can also be removed.

We were surprised by the amount of cervical arthritis for this patient and the image nicely highlights how thoracic hypokyphosis could contribute to cervical degeneration.  Many papers include a clinical example which highlight the conclusions of based on the data presented.  We would like to keep this figure.

Reviewer 2 Report

The manuscript “Cervical Spine Disease Common after Pediatric Treatment of AIS at Mean 30-Year Follow-Up” by  Ernest Young et al. aimed to determine whether sagittal plane alignment affects long-term cervical spine outcomes in AIS patients.

Below are my comments and remarks regarding the manuscript:

1. The term Cervical Osteoarthritis should be used, not arthritis

2. Caption descritpion of table 1 above the table and not below the table

3. Table 2 is too complex, it should be split into separate tables for better readability

4. Table 2 lack of information, the statistical significance of which was compared.

5. A small study group with such a multivariate analysis.

6. With a small study group, why was the multivariate analysis not used?

7. The results of the most important clinical condition of the patient and the correlations resulting therefrom are presented in 4 sentences. There is no table presenting this data. Was radiology assessed or indeed as in Cervical Spine Disease?

8. References should include more recent publications

Author Response

1. The term Cervical Osteoarthritis should be used, not arthritis

Agreed.  Changes made throughout.

2. Caption descritpion of table 1 above the table and not below the table.

Agreed.  Changes made throughout.

3. Table 2 is too complex, it should be split into separate tables for better readability.

Agreed.  This has been done.

4. Table 2 lack of information, the statistical significance of which was compared.

Changes made as above.

5. A small study group with such a multivariate analysis.

Agreed.  However, we have long-term follow-up and we did find some results which were significant.  Thank you for your insight.

6. With a small study group, why was the multivariate analysis not used?

Study was not adequately powered for multivariate analysis.

7. The results of the most important clinical condition of the patient and the correlations resulting therefrom are presented in 4 sentences. There is no table presenting this data. Was radiology assessed or indeed as in Cervical Spine Disease?

We have made multiple changes as suggested to this table.

8. References should include more recent publications

Agreed.  References 31 through 35 have been added.

Reviewer 3 Report

The authors present a long term study (>20 years follow-up) on the impact of AIS treated with either brace or surgery on cervical spine parameters and their correlation with cervical spine degenerative changes. The adult spinal literature has identified sagittal balance as an important predictor of function and AIS patients often have imbalance in their spinal alignment parameters. 

Thoracic hypokyphosis was correlated with degenerative cervical changes. Interestingly, cervical spinal alignment parameters were not correlated with cervical degenerative changes. They found a higher incidence of cervical spine surgery in this population as compared to a population wide prevalence.

These findings suggest that cervical alignment changes are driven by imbalance in the thoracic spine due to their AIS. 

Prior studies on the long term sequelae of AIS have highlighted the impact of alignment on lumbar degeneration. However, this study brings to attention the impact on cervical degeneration as well. Strengths include long term follow-up, however at the consequence of low retention rate and treatment mix.

1) Is there any correlation between lumbopelvic parameters and cervical degeneration?

2) If there are cervical flex-ex films it would be interesting to identify extension and flexion "reserve" available as this has been correlated with neck disability. 

Author Response

Thank you for your positive feedback.  

1)  We were unable to find a correlation between lumbopelvic parameters and cervical arthritis.  Thank you for this interesting question.

2)  Unfortunately, flexion-extension radiographs were not obtained on this population.  Only standing lateral full-length scoliosis x-rays were obtained.  We have added this to our limitations section.

Round 2

Reviewer 1 Report

good

Reviewer 2 Report

I have no further comments, I leave the decision to the Editor